# Contribution from cross-country skiing, start time and shooting components to the overall and isolated biathlon pursuit race performance

Harri Luchsinger, Jan Kocbach, Gertjan Ettema, Øyvind Sandbakk[ID]*

Centre for Elite Sports Research, Department of Neuromedicine and Movement Science, Faculty of Medicine and Health Science, Norwegian University of Science and Technology, Trondheim, Norway

* Oyvind.sandbakk@ntnu.no

## Abstract

### Purpose

Biathlon is an Olympic sport combining 3–5 laps of cross-country skiing with rifle shooting, alternating between the prone and standing shooting positions between laps. The individual distance and the sprint are extensively examined whereas the pursuit, with start times based on the sprint results, is unexplored. Therefore, the current study aimed to investigate the contribution from start time, cross-country skiing time, penalty time, shooting time and range time to the overall and isolated performance in biathlon World Cup pursuit races.

### Methods

38 and 37 stepwise linear regression analyses for each of the races were performed, including 112 and 128 unique athletes where 20 and 13 athletes had more than 20 results within top 30 during the seasons 2011/2012-2015/2016 in men and women, respectively.

### Results

Start time (i.e. sprint race performance) together with penalty time, explained ~80% of the performance-variance ($R^2$) in overall pursuit performance in most races (p<0.01). For isolated pursuit performance, penalty time was the most important component, explaining >54% of the performance-variance in the majority of races, followed by course time (accumulated $R^2$ = .91-.92) and shooting time (accumulated $R^2$ = .98-.99) (p<0.01). Approximately the same rankings of factors were found when comparing standardized coefficients and correlation coefficients of the independent variables included in the regression.

### Conclusion

Start time (i.e. sprint race performance) is the most important component for overall pursuit performance in biathlon, whereas shooting performance followed by course time are the most important components for the isolated pursuit race performance.

**Data Availability Statement:** All analyses were based on publicly available race reports from the International Biathlon Union (IBU) datacenter

(2016), and permission to use the data for scientific purposes was given by the IBU. Link: https://biathlonresults.com/. Permission to use the data for scientific purposes was granted by the IBU secretary general at the time. Due to the data being owned by IBU, other authors are encouraged to contact IBU for the possibility to analyze biathlon race results. Informed consent from the athletes was not necessary to collect, due to the data being publicly available.

**Funding:** The author(s) received no specific funding for this work.

**Competing interests:** The authors have declared that no competing interests exist.

## Introduction

Biathlon is an Olympic sport combining 3–5 laps of cross-country skiing with rifle shooting, alternating between the prone and standing shooting positions between laps. Several different biathlon events exist, in which the individual distance was included as an official World championship-event in 1958, followed by the relay (1960), sprint (1974), pursuit (1997), mass start (1998), mixed relay (2005) and the single mixed relay (2015) [1]. Among the four individual-start formats in biathlon, the individual distance and the sprint are extensively examined, [2–5] whereas the pursuit and the mass start races are almost unexplored [6, 7], although they comprise 50% of the individual-start race formats in the Olympics. In pursuit races, the 60 best athletes from the sprint race chase the leader over 12.5 and 10.0 km for men and women, respectively. The start time in the pursuit race is identical to the result of the sprint race performed 1–3 days before. The pursuit includes two prone and two standing shootings where the penalty loop is the same as for sprint races (150 m/22-24 s for both men and women).

The contribution from the different performance factors in biathlon have been analyzed both for the sprint race and the individual distance. In the sprint, around 60% of the performance difference between those finishing top 10 (G1-10) and those finishing among rank 21–30 (G21-30) was explained by cross-country skiing time (course time) and nearly 40% by shooting performance (i.e. penalty time) in both men and women [5]. The corresponding numbers for the individual distance showed that close to 50% of the overall performance was explained both by cross-country skiing time and shooting performance [3]. These differences between the two disciplines are expected due to the greater penalty for each miss in the individual distance compared to the sprint (i.e. 1 min versus 22–24 s), which is only partly compensated for by the 20% longer lap distance between shootings in the individual distance. In both cases, range time (time on the shooting range when excluding shooting time) and shooting time (time from approaching the shooting mat until the last shot hits the target) explained less than 3% of the performance-difference between G1-10 and G21-30. However, similar analyses for pursuit races do not exist, even though the pursuit differs markedly from other biathlon events since the start time for each athlete is based on the initial sprint race performance. In addition, the pursuit has higher frequency of shootings for each km of skiing compared to other events. The contribution from starting time to the overall performance as an additional main variable may change the impact of cross-country skiing time, shooting performance, shooting time and range time compared to the other events.

In addition, tight duels at the shooting range and the subsequently increased emotional pressure [8] may influence shooting times and range times differently than for races with an interval-start procedure, which could make the shooting component (shooting performance, shooting- and range-time) more important for overall performance and especially for the isolated pursuit performance. The rationale behind this hypothesis is that the shooting component (including shooting time, range time and penalty time) is of higher importance in pursuit races with shorter laps of skiing between shootings than in the sprints and individual distances. In addition, clean shooting and a fast range and shooting time could benefit the cross-country skiing time on the following lap, for example by gained position and positive effects of drafting within a group of athletes. Thus, the understanding of how the main components contribute to overall performance in the pursuit race (including start time/sprint race performance), as well as the contribution of the various components for the isolated pursuit race performance (excluding start time), is of high interest for coaches, athletes, media and the International Biathlon Union (IBU) which governs and organizes international biathlon events.

Therefore, the current study aimed to investigate the contribution from start time, cross-country skiing time, shooting performance, shooting time and range time to the overall and

isolated performance in biathlon World Cup pursuit races in men and women. Due to the impact of start time (i.e. sprint performance) and the high frequency of shootings per distance skied, we hypothesized that start time and penalty time would explain the majority of performance variance in pursuit races for both men and women.

## Methods

This study is based on publicly available race reports and results from the International Biathlon Union (IBU) datacenter (2016), with permission to use the data for scientific purposes given by IBU. A summary of the races included can be found in Table 1.

### Statistical analyses

All statistical analyses were performed using SPSS statistics vs. 23.0, and data were tested for normality using the Shapiro-Wilk test and visual inspection. Data are presented as mean (95% CI).

Stepwise linear regression with total time behind the overall winner (including start time) and total time behind the fastest athlete in the isolated pursuit race (excluding start time) as dependent variables, and course time penalty time, shooting time and range time behind or ahead the overall winner and the fastest athlete in the race as independent variables were performed. The models were applied for top 30 athletes in pursuit races during the seasons 2011/2012-2015/2016. To analyze the importance of the different shootings for the overall penalty time, stepwise linear regression with total penalty time as dependent variable and penalty time from each of the four shootings as independent variables was applied. For the stepwise multiple regressions, outliers and extreme values were defined using boxplots with the range between $1^{st}$ and $3^{rd}$ quartile cutoffs (i.e. 50% of the data lies within the $1^{st}$ and $3^{rd}$ quartile) as reference values. An outlier was defined as being 1.5 times this range away from either of these quartile cutoffs, and extreme values were defined as being more than 3.0 times the range of the $1^{st}$ and $3^{rd}$ quartile-box away from the $1^{st}$ or $3^{rd}$ quartile data-points. This procedure removed 99 outliers or extreme values out of 1140 results among men and 78 out of 1110 results among women, in which five winners and two $2^{nd}$ places were removed from the men's races and 8 winners and three $2^{nd}$ places were removed from the women's races. Removal of the outliers and extreme values only affected the stepwise regressions and correlation analyses and were included for the simple summation of start number and overall rank and the analyzes of overall and isolated pursuit race winners in the results section. Significant multicollinearity between a few independent variables in some of the races were found, but the correlation

**Table 1. Number of races, unique athletes and the average (95% confidence interval) race distance, maximum climb, total climb, air temperature and humidity.**

|  | Men | Women |
|---|---|---|
| Number of races | 38 | 37 |
| Unique athletes | 112 | 128 |
| Unique athletes with >20 results within top 30 | 20 | 13 |
| Race distance (m) | 12740 (12663,12818) | 10396 (10338,10454) |
| Maximum climb (m) | 25 (22,29) | 21 (19,24) |
| Total climb (m) | 83 (80,86) | 64 (60,67) |
| Air temperature (°C) | -0.6 (-2.5,1.4) | -0.6 (-2.5,1.4) |
| Humidity (%) | 70 (64,76) | 70 (63,76) |

Race distance refers to the total distance from start to finish, including the shooting range.

coefficients of these associations were relatively low (mostly 0.3–0.4 and never above 0.6). Although the results of the linear regression analyses must be interpreted with this in mind, we argue that the multicollinearity between independent variables did not affect the conclusions of our study. This is supported by the consistent findings across the various analyses done in our approach.

In addition, independent samples t-tests were used to analyze sex differences in start time, course time, skiing speed, shooting time and numbers of places climbed between men and women both for the overall performance and for time within the isolated pursuit race.

## Results

The average overall racing times (including start time) were 34:20 min (95%CI: 33:50,34:50) and 33:08 min (32:30,33:46), with average isolated pursuit race times of 33:16 min (32:46,33:46) and 31:56 min (31:21,32:32) among top 30 for men and women, respectively. This corresponds to average start times behind the winner of 1:04 min (1:00,1:09) and 1:12 min (1:06,1:17) for men and women, respectively. Out of 20 shots, the average number of misses at the shooting range were 2.6 (2.4,2.8) and 2.8 (2.6,3.1) in each competition among top 30 for men and women, respectively.

### Overall performance

The average total times of the winners were 32:47 min (32:18,33:16) and 30:57 min (30:27, 31:27), with average isolated pursuit race times of 32:35 min (32:06,33:04) and 30:44 min (30:12,31:16) in men and women, respectively.

The overall winner had the fastest race time in the isolated pursuit race in 9% and 13% of the races among men and women, respectively. On average, overall winners started 11.6 s (6.5,16.8) and 13.7 s (8.2,19.3) behind the winner of the sprint in men and women, respectively, with a median start number of 2 among both sexes. In 37% and 32% of the races among men and women, respectively, the overall winner was also the winner of the sprint race. In all except one race, the overall winner started as number 10 or better in both sexes, with 84% and 81% of all victories being achieved by athletes starting as number 5 or better among men and women (Fig 1). However, in 50% of the pursuit races the winner of the sprint ended up more than 51 and 58 seconds behind the overall winner in men and women, respectively, and had the fastest isolated pursuit race time in only one race among both sexes.

Pearson correlation analyses showed that start time correlated most frequently with overall performance in pursuit races (Table 2) followed by penalty time and course time among both men and women.

The results from the stepwise multiple regression analyses are shown in Table 3. The analyses show that start time explained 50–51% of the variance in time behind the overall winner in the 23 and 22 races among men and women, respectively. When additionally including penalty time, the model explained 78–80% of the variance in time behind the overall winner in both sexes.

In addition to the results in Table 3, three races among men and two races among women had best fit for other models with various rankings of the different variables. In one race among men, no variables correlated with overall performance.

The stepwise linear regression with total penalty time as dependent variable showed standing shootings to explain 70–90% of the variance in total penalty time within both sexes, with no difference in the importance from shooting 3 and 4.

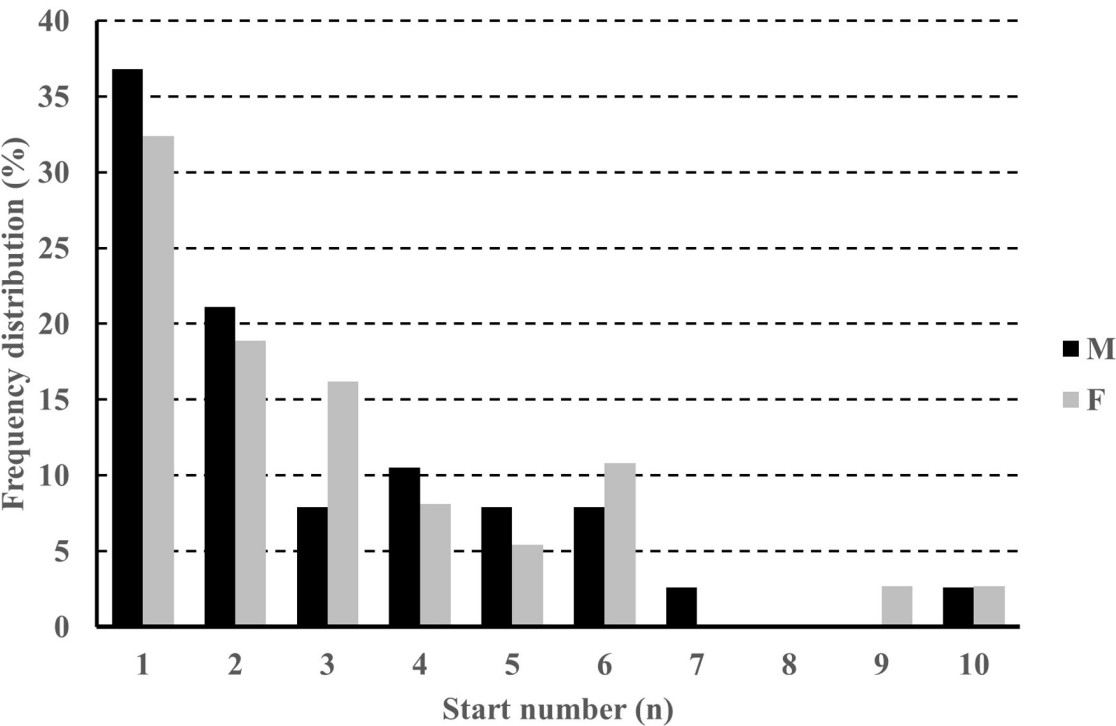

**Fig 1. The distribution of overall pursuit winners in biathlon for the different start numbers in the race (i.e. based on results of the sprint race) in the seasons 2011–2015 in men (M) and women (W).**

### Isolated pursuit race performance

The median start number of athletes having the fastest isolated pursuit race times were 19 and 12, among men and women, respectively. This corresponded to 1:05 min (:54,1:17) and :52 min (:41,1:04) behind the winner of the sprint race and ended up finishing top 5 overall in the pursuit race in 76.3% and 86.5% of the races among men and women, respectively. Here, we found a significant sex difference in start number (p<.05) but not in start time (p = .105). On average, the fastest isolated race time among men gave a final rank [2.9 (2.0,3.8)] closer to the overall victory than among women [4.3 (3.3,5.3), p<.05]. In only 7.9% and 2.7% of the races,

**Table 2. The average correlation coefficients and the number of races with significant positive or negative correlations between time behind the overall pursuit race winner and start, penalty, course, shooting and range time behind the overall winner.**

| Overall pursuit race time behind* | Men | | | | Women | |
|---|---|---|---|---|---|---|
| Variable | Number of positive correlations | Average of the positive correlations | Number of negative correlations | Average of the negative correlations | Number of positive correlations | Average of the positive correlations |
| Start time (s) | 35 | .61 | | | 37 | .64 |
| Penalty time (s) | 35 | .46 | | | 34 | .45 |
| Course time (s) | 26 | .52 | | | 30 | .55 |
| Shooting time (s) | 17 | .42 | | | 7 | .43 |
| Range time (s) | 6 | .36 | 3 | -.38 | 13 | .45 |

*Time behind the overall pursuit race winner was correlated with time behind the overall winner for each of the listed variables. Only significant correlations for each variable were included in the table.

**Table 3. Summary of the stepwise multiple regression analyses performed individually for each race with total time behind the overall winner as dependent variable.**

| | Men | | Women | |
|---|---|---|---|---|
| **Total number of races included** | **38** | | **37** | |
| | **Model outcome 1** | | | |
| Number of races with best fit | 23 | B stand | 22 | B stand |
| 1. Start time | 49.7 (42.8,56.6) | .73 | 50.9 (44.0,57.8) | .64 |
| 2. Penalty time | 79.8 (75.5,84.2) | .68 | 78.1 (74.4,81.9) | .70 |
| 3. Course time | 96.1 (95.4,96.8) | .47 | 95.4 (94.3,96.5) | .54 |
| 4. Shooting time | 99.6 (99.4,99.8) | .22 | 99.8 (99.6,100) | .24 |
| | **Model outcome 2** | | | |
| Number of races with best fit | 7 | B stand | 4 | B stand |
| 1. Penalty time | 40.0 (26.5,53.5) | .84 | 41.3 | .70 |
| 2. Start time | 76.7 (66.9,86.5) | .73 | 73.3 | .54 |
| 3. Course time | 92.0 (85.2,98.8) | .48 | 94.0 | .57 |
| 4. Shooting time | 99.6 (99.1,100.0) | .30 | 99.5 | .24 |
| | **Model outcome 3** | | | |
| Number of races with best fit | 1 | B stand | 5 | B stand |
| 1. Course time | 40.2 | .49 | 50.0 (38.5,61.5) | .59 |
| 2. Penalty time | 73.0 | .59 | 72.8 (66.0,78.7) | .65 |
| 3. Start time | 97.4 | .52 | 95.8 (94.0,97.6) | .55 |
| 4. Shooting time | 99.5 | .18 | 99.8 (99.2,100.0) | .23 |
| | **Model outcome 4** | | | |
| Number of races with best fit | 3 | B stand | 3 | B stand |
| 1. Start time | 55.6 | .72 | 59.6 | .63 |
| 2. Course time | 73.4 | .55 | 78.4 | .57 |
| 3. Penalty time | 94.7 | .63 | 97.4 | .51 |
| 4. Shooting time | 99.4 | .28 | 99.8 | .17 |

Each model lists average cumulated $R^2 * 100$ (including 95% confidence intervals when more than 4 races fit the regression). Start, penalty, course, shooting and range time behind the overall winner were used as independent variables. Each model includes the races where the indicated ranking of the different components [from most (1) to least (4) influential] provided the best fit to the regression. B stand = average of the standardized coefficients.

the athlete with the fastest race time ended up outside of top 10 among men and women, respectively. The average number of misses were lower in men [.79 (.53,1.04)] than in women [1.22 (.93,1.50), $p < .05$], and in 39.5 and 21.6% of the cases, the fastest athlete in the isolated pursuit race missed zero shots, whereas 84.2 and 62.2% hit 19 or 20 out of the 20 shots among men and women, respectively. In addition, 50.0% and 70.3% of the fastest isolated race time-results in men and women, respectively, were among the five fastest in course time in these competitions.

Out of the five main variables, penalty time correlated most strongly with total time behind the fastest isolated race time (Table 4) and correlated significantly with the fastest isolated pursuit race time in all races ($p<.05$).

Results from the stepwise regression analyses, with time behind the fastest isolated pursuit race time as dependent variable, shows that penalty time is the most important component, followed by course time and shooting time in most of the races (Table 5).

In addition to the results in Table 5, two races among women had best fit for models with other rankings of the variables.

**Table 4. The average correlation coefficients and the number of races with significant positive or negative correlations between time behind the fastest isolated pursuit race time and start, penalty, course, shooting and range time behind the athlete with the fastest isolated pursuit race time.**

| Isolated pursuit race time behind* | Men | | | | Women | | | |
|---|---|---|---|---|---|---|---|---|
| Variable | Nr. of positive correlations | Avrg. of the positive correlations | Nr. of negative correlations | Avrg. of the negative correlations | Nr. of positive correlations | Avrg. of the positive correlations | Nr. of negative correlations | Avrg. of the negative correlations |
| Penalty time* (s) | 38 | .76 | | | 37 | .68 | | |
| Course time* (s) | 28 | .51 | | | 30 | .51 | | |
| Start time* (s) | 1 | .35 | 11 | -.44 | 6 | .40 | 3 | -.41 |
| Shooting time* (s) | 12 | .44 | | | 7 | .43 | | |
| Range time* (s) | 1 | .32 | 1 | -.36 | 6 | .40 | 1 | -.32 |

*Time behind the fastest athlete in the isolated pursuit race was correlated with the time behind the athlete with the fastest isolated pursuit race time for each of the listed variables. Only significant correlations for each variable were included in the table.

## Discussion

This study investigated the contribution from start time, cross-country skiing performance and shooting performance in biathlon World Cup pursuit races, as well as these factors' importance to isolated pursuit race performance. The main findings show that in 60% of the races, start time (i.e. sprint race performance) was the most important component, explaining approximately 50% of the variance in overall performance among both men and women. This was followed by penalty time, which together with start time explained approximately 80% of the overall performance in both sexes. When further adding course time in the regression

**Table 5. Summary of the stepwise multiple regression analyses performed individually for each race with total time behind the isolated pursuit race winner as dependent variable.**

| | Men | | Women | |
|---|---|---|---|---|
| Total number of isolated pursuit race performances included | 38 | | 37 | |
| | | | Model outcome 1 | |
| Number of races with best fit | 35 | B stand | 27 | B stand |
| 1. Penalty time | 61.7 (57.4,66.0) | .87 | 54.1 (49.2,59.0) | .90 |
| 2. Course time | 91.7 (90.5,93.0) | .59 | 91.1 (89.4,92.8) | .70 |
| 3. Shooting time | 99.0 (98.8,99.3) | .29 | 99.3 (99.0,99.6) | .31 |
| 4. Range time | 100 | .11 | 100 | .09 |
| | | | Model outcome 2 | |
| Number of races with best fit | 3 | B stand | 8 | B stand |
| 1. Course time | 45.0 | .80 | 44.1 (33.3,55.0) | .85 |
| 2. Penalty time | 91.7 | .92 | 92.0 (88.6,95.4) | .84 |
| 3. Shooting time | 98.3 | .32 | 99.1 (98.3,100.0) | .30 |
| 4. Range time | 100 | .14 | 100 | .10 |

Each model lists average cumulated $R^2$*100 (including 95% confidence intervals when more than 4 races fit the regression). Penalty, course, shooting and range time behind the isolated pursuit race winner were used as independent variables. Each model includes all races where the indicated ranking of the different components [from most (1) to least (4) influential] fit the model best. B stand = average of the standardized coefficients.

analyses, the model explained 95–96% of the variance in overall performance in both men and women. In addition, analyses of the isolated pursuit race performance showed that in 92 and 73% of the races among men and women, respectively, penalty time was the most important component followed by course time and shooting time, explaining >54, 91–92 and 98–99% of the performance-variance. Both for overall and isolated pursuit race performance, approximately the same rankings of factors were found when comparing standardized coefficients and correlation coefficients of the independent variables included in the regression.

## Overall performance

Our analyses show that start time, that is sprint race performance, is the most important component for the overall pursuit race performance. Above 80% of the overall winners started as number 5 or better after the sprint among both men and women, and the regression analyses show that in 23 and 22 races out of the 38 and 37 pursuit races investigated in men and women, respectively, 50% of the overall performance is explained by start time. Altogether this highlights the importance of the sprint race to the overall pursuit race performance in biathlon.

Penalty time was ranked as the second most contributing component in 23 and 22 races of the pursuit races. Regression analyses showed that start time and penalty time together explained approximately 80% of the overall performance in these races. In 7 and 4 races among men and women, respectively, penalty time was ranked as the most important component, with regression analyses showing that approximately 40% of the overall pursuit performance variance was explained by penalty time in both men and women. Our findings also show that winners of pursuit races very rarely have more than 2 misses, that mostly occur in the standing shootings which also explains most of the variance in penalty time. In addition, there was no sex difference in penalty time among top 30 athletes. This is in line with previous findings in sprint showing that top 10-athletes in sprint races on average hit more than 90% of the targets, where most of the misses occur during standing shooting and that there is no sex difference in shooting performance within top 30 [5]. Together with the large standardized coefficients and high frequency of significant correlations between penalty time and overall performance, this emphasizes the importance of the shooting component and especially performance in the standing shootings to overall pursuit race performance.

Course time was the third most important component in most of the pursuit races, where the regression analyses showed that the model increased its explanatory fit from approximately 80% with start and penalty time included in the model, to more than 95% when course time was included. The relatively low importance of course time compared to start time and penalty time might be explained by the advantage of skiing in a group, because of drafting that is often the case in pursuit races. This would logically make the start time and penalty time more important since athletes who are originally faster skiers have difficulties breaking away from a group and slower skiers can join groups of skiers that are normally faster in individual-start races. In addition, the athletes starting early in the pursuit race might use a more conservative pacing strategy to prepare for shooting in the beginning of the race compared to those chasing from behind. This corresponds with more even pacing, as shown previously for better performing athletes in biathlon sprint races [9].

Shooting time was ranked as the fourth most contributing component in almost all races, explaining on average 3–7% of the performance-variance. This is more than previously found for the sprint and individual distance, which makes sense because the frequency of shootings relative to the skiing distance in pursuits is higher [10]. Furthermore, fast shooting probably provides an advantage in duel shooting to climb places compared to events with interval-start

procedure. In their review of the scientific literature in biathlon, together with analyses of the Olympic biathlon events in Pyeongchang, Laaksonen et al. [10] suggested that fast and clean shooting (no mistakes) would become even more important to win future biathlon races.

Range time contributed significantly to the overall performance in only one of the 38 races among men and in none of the races among women. This is in contrast to research from 1992, that indicated that biathletes could save approximately 10 s in range time by maintaining speed in the last 50 m before shooting [11]. This is no longer the case either in the sprint [5], individual [3], and according to the present results, in pursuit races.

## Isolated pursuit race performance

Since start time (i.e. the previous sprint race performance) explains 50% of the variance in overall performance within both men and women in most of the races, it is of further interest to understand how the different components contribute to the isolated pursuit race (i.e. when excluding start time). Our analyses show that penalty time is the most important component for the isolated pursuit race performance in almost all races among men and in around 80% of races among women, explaining approximately 62 and 54% of the variance in race time in men and women, respectively.

Course time was the second most important component for the isolated pursuit race performance, which together with penalty time explains more than 90% of the performance-variance in isolated pursuit races. The fastest isolated pursuit race times among women are to a greater extent than among men explained by faster skiing and to a lesser extent by shooting performance. This indicates a greater opportunity for faster skiers in the women's class to climb ranks in the pursuit race.

Shooting time was more important for the isolated pursuit race performance than for the overall pursuit race performance, explaining approximately 8% of the variance in isolated pursuit race time in both men and women. This means that shooting time is an important component for the isolated pursuit race performance. Together, the importance of penalty time and shooting time highlights the high importance of the shooting component for the isolated pursuit race performance, as it explains approximately 60–70% of the performance-variance in both sexes. In addition, the fastest athletes in the isolated pursuit race among women tended to shoot slower than men, in line with previous research on the sprint and individual distances [3, 5, 10, 12], indicating that there is more to gain in shooting time among women than among men.

Start time correlated negatively with isolated pursuit performance in 11 races among men and in 3 races among women, which suggests that start time provides a larger advantage for women than for men. This could be related to the larger time-gap between athletes after the sprint race in the women's class compared to men.

The size of the standardized coefficients in the regression analyses and the frequency and strength of significant correlations between the various independent variables and pursuit performance shows a similar picture as the regression analyses. Although this study indicates that shooting is more important in pursuits than in sprint races, start time explains a large portion of performance in biathlon pursuit races. Thus, the same components as for the sprint distance should also be emphasized when training for the pursuit. However, our analyses show that the fastest athletes in the isolated pursuit race, started on average as number 20 and 14 and ended up finishing top 5 overall in 76 and 87% of the races among men and women, respectively. In addition, the winner of the sprint race rarely had the fastest isolated pursuit race time and in half of the races ended up approximately 1 minute behind the overall winner. Furthermore, penalty time explains most of the variance for the isolated pursuit race result in most of the

races in both sexes. In addition, most of the variance in penalty time was explained by the two last shootings in pursuit races for both sexes. Therefore the uncertainty in outcome, which is important in competitive sports [13], is maintained until the last shootings in the pursuit in biathlon. This factor has likely also contributed to the increase in popularity of biathlon [13], with a race format leading to tight duels at the shooting range where the first athlete to cross the finish line is the overall winner. While the same factors generally contribute to performance in both sexes, the current and previous results indicate that coaches and athletes should be aware of the different performance demands in the men's and women's class and especially consider the possibility for shooting faster among women.

## Methodological considerations

We argue that the analyses of all 38 and 37 races provides a good overall picture on the most important race components contributing to overall and isolated pursuit race performance. However, the effect of course profile, weather conditions and other factors such as mental pressure in Championships would be logical explanatory factors for the within-race differences that should be considered when analyzing single races.

For the stepwise regression analyses, each race was analyzed individually and for this reason the model outcomes cannot be generalized to all races. However, supporting the stepwise regression analyses employed here, our analyses of standardized coefficients together with the simple descriptive statistics and correlational analyses supported the main findings outlined. Thus, we argue that these findings together provide a comprehensive picture of the importance of cross-country skiing, start time and shooting components to the overall and isolated biathlon pursuit race performance.

Significant multicollinearity between a few independent variables in some of the races were found, but the correlation coefficients of these associations were relatively low (mostly 0.3–0.4 and never above 0.6). Although the results of the linear regression analyses must be interpreted with this in mind, we argue that the multicollinearity between independent variables did not affect the conclusions of our study. This is supported by the consistent findings across the various analyses done in our approach.

Shooting times are extracted from the range times based on the manual recordings of shooting time and shooting time and range time data are therefore not highly accurate. However, this error is random and unlikely to influence the conclusions in our approach. Still, some caution should be made when interpreting the results of the present study.

## Conclusions

Start time is the most important component for overall pursuit performance in biathlon, demonstrating that performance in the preceding sprint race is the most important component in the biathlon pursuit. This is followed by penalty time as the second most contributing component, which together with start time explain approximately 80% of the variance in overall pursuit race performance in both men and women. When excluding start time, penalty time is the most important component of the isolated pursuit race performance in almost all races among men and in most races for women, with course time being the second most important component.

## Supporting information

**S1 File.**
(XLSX)

## Acknowledgments

The authors thank IBU for their willingness to share competition data.

## Author Contributions

**Conceptualization:** Harri Luchsinger, Jan Kocbach, Gertjan Ettema, Øyvind Sandbakk.

**Data curation:** Harri Luchsinger, Jan Kocbach.

**Formal analysis:** Harri Luchsinger, Jan Kocbach, Gertjan Ettema, Øyvind Sandbakk.

**Methodology:** Harri Luchsinger, Jan Kocbach, Gertjan Ettema, Øyvind Sandbakk.

**Project administration:** Harri Luchsinger, Øyvind Sandbakk.

**Supervision:** Jan Kocbach, Gertjan Ettema, Øyvind Sandbakk.

**Visualization:** Harri Luchsinger, Jan Kocbach.

**Writing – original draft:** Harri Luchsinger, Øyvind Sandbakk.

**Writing – review & editing:** Harri Luchsinger, Jan Kocbach, Gertjan Ettema, Øyvind Sandbakk.

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
