## [Decision Letter · Decision Letter 0]

19 Mar 2020

PONE-D-20-03338

Contribution from cross-country skiing, start time and shooting components to the overall and isolated biathlon pursuit race performance

PLOS ONE

Dear Dr Sandbakk,

Thank you for submitting your manuscript to PLOS ONE. After careful consideration, we feel that it has merit but does not fully meet PLOS ONE’s publication criteria as it currently stands. Therefore, we invite you to submit a revised version of the manuscript that addresses the points raised during the review process.

Manuscripts needs deep revising. Please, address issues raised by Reviewers 1 and 2.

We would appreciate receiving your revised manuscript by May 03 2020 11:59PM. To enhance the reproducibility of your results, we recommend that if applicable you deposit your laboratory protocols in protocols.io, where a protocol can be assigned its own identifier (DOI) such that it can be cited independently in the future. For instructions see: http://journals.plos.org/plosone/s/submission-guidelines#loc-laboratory-protocols

We look forward to receiving your revised manuscript.

Kind regards,

Luca Paolo Ardigò, Ph.D.

Academic Editor

PLOS ONE

Journal Requirements:

Additional Editor Comments:

Manuscripts needs deep revising. Please, address issues raised by Reviewers 1 and 2.

Reviewers' comments:

Reviewer's Responses to Questions

**Comments to the Author**

1. Is the manuscript technically sound, and do the data support the conclusions?

Reviewer #1: Yes

Reviewer #2: Partly

Reviewer #3: Yes

2. Has the statistical analysis been performed appropriately and rigorously? 

Reviewer #1: Yes

Reviewer #2: No

Reviewer #3: Yes

3. Have the authors made all data underlying the findings in their manuscript fully available?

Reviewer #1: Yes

Reviewer #2: Yes

Reviewer #3: Yes

4. Is the manuscript presented in an intelligible fashion and written in standard English?

Reviewer #1: Yes

Reviewer #2: Yes

Reviewer #3: Yes

5. Review Comments to the Author

Reviewer #1: The authors present interesting data about how different components affect the biathlon pursuit performance. The paper is mostly clearly written, relatively easy to follow and is likely of importance for coaches and athletes in the biathlon sport. However, in overall, the results section is a bit messy, sometimes hard to follow.

Specific comments:

Abstract

L38 - standing shooting positions.

L41 - shooting performance. This includes shooting time and accuracy. I would use the penalty time here as well as you have done in the main text.

L46 - Start time (i.e. sprint race performance)

L46 - explainED 80%.

L53 - pursuit race PERFORMANCE.

Introduction

Mostly clearly written and logical. Comments:

L59 - prone and standing shooting positions

L59-62 - Yes, biathlon is an Olympic winter sport but here it is a little bit confusing as e.g. single-mixed relay is not included in the Olympic program. Revise this sentence.

L63 - I would add Dzhilkibaeva et al. 2018 here as well.

L79 - 20% longer lap distance between shots?? Do yo mean shootings? Or Shooting stations?

L93 - In addition, good shooting and... what is meant by good shooting? Fast, clear or both?

L99 - just a curiosity - how could organizers benefit from the results of this study?

L100-105 - You should define the shooting time and range time here (or in the Introduction somewhere).

Methods

L107-114 - Add information which seasons were included in the analysis (this information is in the abstract)

L123 - range time excluding shooting time. This is sensitive for the outcome of your study. As shooting time is taken manually, there is a source of error. Let say, a true shooting lasts 25 seconds, +/- 2 seconds error in manual measurement means almost 10% error. When you also extract this value from the range time which is "correctly" measured using the timing-chip system, you will get a double error (error in range time as well). I understand this approach from regression analysis point of view, but you must at least mention this complex of problems in the discussion.

Results

Interesting you point out that basically you must start within top 5, at least top 10, to have a chance to win the pursuit race. Good! However, consider revising the text in the results section. It is sometimes difficult to follow and one must read the same sentence a couple of times to understand what do you mean.

L157 - remove (bib1)

L164 - women (F) - should be women (W) (also in the figure 1)

L165 - Pearson correlation; add 'analysis'

Discussion

L211 - maybe add "in both sexes"

L227-230 - add 'in sprint'

L233-L240 - maybe add something about pacing & drafting here.

L243 - frequency of shootings? frequency of shooting tasks?

L252-253 - maybe revise to "...(5), individual (3), and according to the present results, in pursuit races.'

L273 - 60-70% in both sexes?

L279 - bigger - change to larger.

L287-288 - Furthermore, penalty time... of the variance in the isolated pursuit race result. You have not shown data for this conclusion. You have presented that shooting 3 and 4 explained 70-90% of the variance in total penalty time (L177).

Reviewer #2: The paper proposes an approach to evaluate the impact of several explanatory variables, like start and penalty time, on overall and isolated biathlon pursuit performance. The authors use stepwise linear regressions to identify the most important predictors. Furthermore, they use correlation analyses to examine the linear relationships between start time (penalty time, course time, shooting time, range time) and overall (isolated) pursuit race time behind. Particularly, they focus also on the differences between male and female biathletes. Additionally, they describe the relationship between sprint and pursuit races.

In general, the paper discusses an interesting topic but it partially lacks of correct descriptions and applications of statistical methods. Until now, the manuscript needs to be revised because there are fundamental issues which undermine the quality of the paper severely. Furthermore, the interpretation and discussion of the results should be expanded. I recommend a major revision of the manuscript.

You can find the major and minor comments in the attached file.

Reviewer #3: The article submitted for review covers important and current issues of determinants of sport results. Structure of the work is correct. It includes original research problem, which is characterized properly and transparently. Statistical methods chosen well. The work is written well.

6. PLOS authors have the option to publish the peer review history of their article (what does this mean?). If published, this will include your full peer review and any attached files.

Reviewer #1: No

Reviewer #2: No

Reviewer #3: No

---

## [Author Response · Author response to Decision Letter 0]

15 May 2020

Please see attached document with responses to the reviewers

---

## [Decision Letter · Decision Letter 1]

22 Jun 2020

PONE-D-20-03338R1

Contribution from cross-country skiing, start time and shooting components to the overall and isolated biathlon pursuit race performance

PLOS ONE

Dear Dr. Sandbakk,

Thank you for submitting your manuscript to PLOS ONE. After careful consideration, we feel that it has merit but does not fully meet PLOS ONE’s publication criteria as it currently stands. Therefore, we invite you to submit a revised version of the manuscript that addresses the points raised during the review process.

Please, one further effort to address further minor revisions requested by reviewers.

We look forward to receiving your revised manuscript.

Kind regards,

Luca Paolo Ardigò, Ph.D.

Academic Editor

PLOS ONE

Additional Editor Comments (if provided):

Please, one further effort to address further minor revisions requested by reviewers.

Reviewers' comments:

Reviewer's Responses to Questions

**Comments to the Author**

1. If the authors have adequately addressed your comments raised in a previous round of review and you feel that this manuscript is now acceptable for publication, you may indicate that here to bypass the “Comments to the Author” section, enter your conflict of interest statement in the “Confidential to Editor” section, and submit your "Accept" recommendation.

Reviewer #1: All comments have been addressed

Reviewer #2: All comments have been addressed

2. Is the manuscript technically sound, and do the data support the conclusions?

Reviewer #1: Yes

Reviewer #2: Partly

3. Has the statistical analysis been performed appropriately and rigorously? 

Reviewer #1: Yes

Reviewer #2: N/A

4. Have the authors made all data underlying the findings in their manuscript fully available?

Reviewer #1: Yes

Reviewer #2: Yes

5. Is the manuscript presented in an intelligible fashion and written in standard English?

Reviewer #1: Yes

Reviewer #2: Yes

6. Review Comments to the Author

Reviewer #1: I'm happy with the revision the authors have made. I have only one minor issue:

L343: "Extracting the shooting time from Range time" You propose that the error marginal is 1 sec but when thinking of pursuit competition and within the races you have been studying, there is definitively several cases when there are quite a many biathletes approaching and/or leaving the shooting line at the same time. This means that it is impossible for the volunteers to push "several buttons" at the same time. I agree that the error is random, but I suggest to revise the last sentence in methodological considerations. Something with style: "However, the error is random but this possible error needs to be considered when interpreting the of the present study."

Reviewer #2: First of all, I want to thank the authors for the detailed answers to the comments of the first review. The authors have revised their manuscript precisely. Nonetheless, there is a major limitation of the analysis and it would be appreciated if the authors would add the limitation in their manuscript for publication. The added methodological considerations section is a first step to pick up the limitations of the analysis. But I think one major limitation, which was also mentioned in the first review, should be clearly communicated to the reader. As for each race, an individual regression model is fitted, the results are only valid for this race. Thus, no generalization on the basis of the regression results holds, which means that the results should be interpreted conservatively. Your answer in the first review is correct: ''As the results are provided now, the reader can evaluate if 23/38 races with start time being the most important factor among men is many or few races. If a more general result was to be provided, the results in model outcome 2 would affect the results in model outcome 1 and vice-versa.". For this reason, I think an additional paragraph should be added to the methodological considerations section that clarifies the limitation and ensures that the reader doesn't make wrong conclusions.

Additionally, the independent variables are likely to have high correlations (although no multicollinearity statistics are provided), which could bias the results. Thus, it would not clear which variable contributes to the dependent variable. This could invalidate the statements on the importance of certain variables. For this reason, it is necessary to check if this problem arises. Should this be the case then the authors have to find a solution to this problem because as mentioned above it would lead to wrong conclusions. Otherwise, they should emphasize that no problem on the basis of correlation exists.

To sum up, I appreciate the changes in the manuscript. The authors should clearly mention the problem of generalization in their manuscript and they should check if the problem of high correlations between the independent variables exists. For this reason, I recommend a minor revision if the problem of high correlations does not occur.

7. PLOS authors have the option to publish the peer review history of their article (what does this mean?). If published, this will include your full peer review and any attached files.

Reviewer #1: No

Reviewer #2: No

---

## [Author Response · Author response to Decision Letter 1]

5 Aug 2020

Reviewer #1: I'm happy with the revision the authors have made. I have only one minor issue:

L343: "Extracting the shooting time from Range time" You propose that the error marginal is 1 sec but when thinking of pursuit competition and within the races you have been studying, there is definitively several cases when there are quite a many biathletes approaching and/or leaving the shooting line at the same time. This means that it is impossible for the volunteers to push "several buttons" at the same time. I agree that the error is random, but I suggest to revise the last sentence in methodological considerations. Something with style: "However, the error is random but this possible error needs to be considered when interpreting the of the present study."

Response: Thank you for your advice upon this subject. We have now adjusted the last sentence in the methodological considerations paragraph according to the suggestion provided. Please see changes marked in yellow in the revised manuscript. We kept the previous changes in red color.

Reviewer #2: First of all, I want to thank the authors for the detailed answers to the comments of the first review. The authors have revised their manuscript precisely. Nonetheless, there is a major limitation of the analysis and it would be appreciated if the authors would add the limitation in their manuscript for publication. The added methodological considerations section is a first step to pick up the limitations of the analysis. But I think one major limitation, which was also mentioned in the first review, should be clearly communicated to the reader. As for each race, an individual regression model is fitted, the results are only valid for this race. Thus, no generalization on the basis of the regression results holds, which means that the results should be interpreted conservatively. Your answer in the first review is correct: ''As the results are provided now, the reader can evaluate if 23/38 races with start time being the most important factor among men is many or few races. If a more general result was to be provided, the results in model outcome 2 would affect the results in model outcome 1 and vice-versa.". For this reason, I think an additional paragraph should be added to the methodological considerations section that clarifies the limitation and ensures that the reader doesn't make wrong conclusions.

Additionally, the independent variables are likely to have high correlations (although no multicollinearity statistics are provided), which could bias the results. Thus, it would not clear which variable contributes to the dependent variable. This could invalidate the statements on the importance of certain variables. For this reason, it is necessary to check if this problem arises. Should this be the case then the authors have to find a solution to this problem because as mentioned above it would lead to wrong conclusions. Otherwise, they should emphasize that no problem on the basis of correlation exists.

To sum up, I appreciate the changes in the manuscript. The authors should clearly mention the problem of generalization in their manuscript and they should check if the problem of high correlations between the independent variables exists. For this reason, I recommend a minor revision if the problem of high correlations does not occur.

Response: Thank you for your thorough reply and positive criticism of this manuscript. We have now added additional information about the implications of analyzing each race individually in the methodological considerations section in the revised manuscript. Please see changes marked in yellow in the revised manuscript. We kept the previous changes marked in red color.

We found significant multicollinearity between a few independent variables in some of the races, but the correlation coefficients of these associations were relatively low (mostly 0.3-0.4 and never above 0.6). Although the results of the linear regression analyses must be interpreted with this in mind, we argue that this multicollinearity between independent variables did not affect the conclusions of our study. This is supported by the consistent findings across the various analyses done in our approach. We added a paragraph about this in the methodological consideration section.

---

## [Decision Letter · Decision Letter 2]

31 Aug 2020

Contribution from cross-country skiing, start time and shooting components to the overall and isolated biathlon pursuit race performance

PONE-D-20-03338R2

Dear Dr. Sandbakk,

We’re pleased to inform you that your manuscript has been judged scientifically suitable for publication and will be formally accepted for publication once it meets all outstanding technical requirements.

Kind regards,

Luca Paolo Ardigò, Ph.D.

Academic Editor

PLOS ONE

Additional Editor Comments (optional):

Congratulations for your interesting study.

Reviewers' comments:

Reviewer's Responses to Questions

**Comments to the Author**

1. If the authors have adequately addressed your comments raised in a previous round of review and you feel that this manuscript is now acceptable for publication, you may indicate that here to bypass the “Comments to the Author” section, enter your conflict of interest statement in the “Confidential to Editor” section, and submit your "Accept" recommendation.

Reviewer #1: All comments have been addressed

Reviewer #2: All comments have been addressed

2. Is the manuscript technically sound, and do the data support the conclusions?

Reviewer #1: Yes

Reviewer #2: Yes

3. Has the statistical analysis been performed appropriately and rigorously? 

Reviewer #1: Yes

Reviewer #2: Yes

4. Have the authors made all data underlying the findings in their manuscript fully available?

Reviewer #1: Yes

Reviewer #2: Yes

5. Is the manuscript presented in an intelligible fashion and written in standard English?

Reviewer #1: Yes

Reviewer #2: Yes

6. Review Comments to the Author

Reviewer #1: I would like to thank the authors for a great work with this manuscript which hopefully will help the coaches and athletes in biathlon.

Reviewer #2: I am very appreciated that the authors add my previous comments to their methodological considerations. Finally, I have no further review comments and I recommend to accept the manuscript for publication.

7. PLOS authors have the option to publish the peer review history of their article (what does this mean?). If published, this will include your full peer review and any attached files.

Reviewer #1: No

Reviewer #2: No

---

## [Editor Report · Acceptance letter]

4 Sep 2020

PONE-D-20-03338R2 

Contribution from cross-country skiing, start time and shooting components to the overall and isolated biathlon pursuit race performance 

Dear Dr. Sandbakk:

I'm pleased to inform you that your manuscript has been deemed suitable for publication in PLOS ONE. Congratulations! Your manuscript is now with our production department. 

Kind regards, 

on behalf of

Dr. Luca Paolo Ardigò 

Academic Editor

PLOS ONE